# Genetic Influences on Fetal Alcohol Spectrum Disorder

**DOI:** 10.3390/genes14010195

**Published:** 2023-01-12

**Authors:** Danielle Sambo, David Goldman

**Affiliations:** National Institute on Alcohol Abuse and Alcoholism, 5625 Fishers Lane, Rockville, MD 20852, USA

**Keywords:** prenatal alcohol exposure, fetal alcohol spectrum disorder, genetics, alcohol dehydrogenase, genetic testing

## Abstract

Fetal alcohol spectrum disorder (FASD) encompasses the range of deleterious outcomes of prenatal alcohol exposure (PAE) in the affected offspring, including developmental delay, intellectual disability, attention deficits, and conduct disorders. Several factors contribute to the risk for and severity of FASD, including the timing, dose, and duration of PAE and maternal factors such as age and nutrition. Although poorly understood, genetic factors also contribute to the expression of FASD, with studies in both humans and animal models revealing genetic influences on susceptibility. In this article, we review the literature related to the genetics of FASD in humans, including twin studies, candidate gene studies in different populations, and genetic testing identifying copy number variants. Overall, these studies suggest different genetic factors, both in the mother and in the offspring, influence the phenotypic outcomes of PAE. While further work is needed, understanding how genetic factors influence FASD will provide insight into the mechanisms contributing to alcohol teratogenicity and FASD risk and ultimately may lead to means for early detection and intervention.

## 1. Introduction

Prenatal alcohol exposure (PAE) is linked to miscarriage, stillbirth, preterm birth, and sudden infant death syndrome [1] and is characterized by several other deleterious outcomes in surviving offspring, including growth restriction, morphological abnormalities, cognitive dysfunction, and behavioral issues. Collectively, the range of adverse effects of PAE is encompassed under the umbrella term fetal alcohol spectrum disorder (FASD), for which four major diagnostic subcategories have been described: FAS, pFAS, ARND, and ARNB. The most severe form of FASD is fetal alcohol syndrome (FAS), with diagnosis requiring (A) characteristic FAS facial dysmorphology (short palpebral fissures, thin vermillion, and smooth philtrum), (B) growth restriction, (C) deficient growth or abnormal development of the brain, and (D) neurobehavioral impairment. While FAS does not require documented PAE, partial FAS (pFAS) can be diagnosed with confirmed PAE, the FAS facial phenotype, and growth restriction (features A and B), or without confirmed PAE and features A, B, and deficient brain development (feature C). Diagnosis of alcohol-related neuronal disorder (ARND) requires documented PAE and neurobehavioral impairment, while the diagnosis of alcohol-related birth defects (ARNB) requires documented PAE and one or more specific malformations known to be consequent to PAE [2].

Although public health guidelines continue to discourage alcohol use during pregnancy, PAE is the leading cause of preventable developmental disability worldwide. The estimated incidence of FASD is high both in the United States (10 per 1000 children [3]) and worldwide (8 per 1000 children [4]) and astoundingly high in several countries, with a meta-analysis revealing the highest rates in South Africa (111 per 1000), Croatia (53 per 1000), and Ireland (48 per 1000) [4]. Various problems with diagnosis, however, blur the relationship between PAE and FASD. Lack of confirmed PAE is a significant barrier to diagnosis, with documented PAE being a criterion of three of the four FASD diagnostic categories. The absence of information on maternal use of alcohol is especially problematic in foster and adopted children who are 10–15 times more likely to have FASD but in whom PAE may be unconfirmed [5]. Even when the mother is available to provide a history, due to the stigma surrounding alcohol and drug use during pregnancy, PAE is likely to be significantly underreported, if not denied completely. Furthermore, the FASD diagnosis remains challenging. Although the diagnostic criteria for FASD have been thoroughly described, specialized training is required; assessment techniques are still debated, especially for ARND and ARNB; and children may have other disorders or secondary disabilities that make it difficult to isolate FASD [6,7]. Taken together, the interrelated problems of PAE measurement and diagnosis of FASD contribute to the underdiagnosis of FASD. In an analysis of 6639 children across four different communities in the United States, 222 children (1 in 30) were diagnosed with FASD, of whom only 2 were previously diagnosed [3].

Despite the high prevalence of FASD, which is likely underestimated, most children exposed to alcohol prenatally do not develop FASD [8]. Studies suggest that only 4–10% of children with heavy PAE develop full FAS [9,10], and only 10–15% of children with any PAE develop more broadly defined FASD [11]. Several factors contribute to the risk of FASD and its severity. The timing, dose, frequency, and duration of maternal drinking influence the type and intensity of outcomes. Higher doses, more frequent use, and prolonged use of alcohol during pregnancy increases the likelihood of more severe effects, while the timing of exposure at different developmental stages influences the type of physical and neurobehavioral effects. Although the pattern of PAE greatly contributes to FASD risk and severity, other maternal factors also lead to variation in FASD severity. Higher maternal age, gravidity, and parity are all associated with more severe FASD outcomes compared to women with similar drinking patterns [12]. Malnutrition or deficiencies in select micronutrients also increase the risk of FASD, especially in populations otherwise at high risk [13]. In that regard, the risk of conceiving a child with FAS is 15.8 times higher for women of lower socioeconomic status even when alcohol intake is comparable [14]. Importantly, several of the maternal risk factors for FASD are also associated with lower birth weight, malformations, and attention deficit disorder in general [12], and PAE is believed to exacerbate these effects.

While FASD essentially is contingent on exposure, genotypes of both mother and offspring appear to also influence FASD and the severity of alcohol teratogenicity, shifting reaction range to exposure. The role of genetic influences in FASD is known from preclinical studies, with different strains of mice, rats, chickens, and fish having been shown to have different susceptibilities to PAE effects [15]. For example, in 11 different chicken strains, chick embryos could be classified as very sensitive, moderately sensitive, or insensitive to PAE [16]. Several reports show that the C57BL/6J mouse strain has a higher susceptibility to alcohol teratogenicity compared to the CBA/2J strain as well as the C57BL/6N sub-strain [17,18]. In this article, we review the genetics of FASD in humans, considering both the inheritance of FASD and genes linked to FASD susceptibility in a variety of populations, recognizing that genetic variations altering response to PAE may not be shared with other species that represent complementary models. While the genetics of FASD has been reviewed [15,19,20], here, we focus on human FASD cohorts and populations, including twin studies, candidate gene studies, and genetic testing. Genes of both the mother and offspring may influence vulnerability to FASD by altering maternal alcohol consumption, alcohol metabolism, uterine environment, or fetal growth or by a combination of pathways. While FASD genetics is poorly understood, these studies inform us that genetic influences contribute to FASD etiology and provide potential tools for diagnosis and intervention.

## 2. Discordance and Concordance in Twin Studies

Some of the most compelling evidence for genetic influences on FASD susceptibility and severity is derived from twin studies, small and few that these studies have been. Early case studies in dizygotic (DZ) twins born to mothers with moderate to high alcohol use during pregnancy led to the knowledge that one twin may often be more affected than the other [21,22,23,24]. In 1993, a seminal study of monozygotic (MZ) and DZ twins found that in 5 MZ and 11 DZ twin pairs born to “alcohol-abusing” mothers, all MZ twins were concordant for diagnosis of FAS or fetal alcohol effects (FAE), whereas 7 of 11 DZ twins (64%) were concordant [25]. In two of the four discordant DZ twins, one of the twin pairs had no FAS or FAE diagnosis. Of note, this early study of 16 twin pairs and the other studies of single twin pairs were conducted before the creation of the more rigorous FASD diagnostic system, and therefore only two FASD diagnoses (FAS and FAE) were applied to the children.

A more recent, although similarly small, study from the Fetal Alcohol Syndrome Diagnostic and Prevention Network also supports high heritability for FASD [26]. In addition to MZ and DZ twins, this study included full-siblings and half-siblings with a common birth mother. The sibling pairs were obviously not simultaneously exposed but were reported to have had similar PAE and other maternal risk factors. Like the Streissguth and Dehaene study, all 9 MZ twins in this study were concordant for FASD, while the 39 DZ twins were 56% concordant. In four of the DZ pairs, one twin presented with severe dysfunction while the other twin pair had low to moderate effects. In the 24 full-sibling pairs and 9 half-sibling pairs, concordance was 41% and 22%, respectively. Thus, the pair-wise concordances of these pairs of relatives mirrored their coefficients of relationship, with MZ twins, DZ twins, full-siblings, and half-siblings being approximately 100, 50, 50, and 25% concordant, respectively. These concordances can be used to calculate a heritability of 100%, although with a large standard deviation because of the small number of pairs of relatives studied. These findings suggest that with virtually identical PAE, in utero environment, and maternal risk factors in twins, as well as similar exposures and risk factors in siblings, differences in vulnerability to the teratogenic effects of alcohol are strongly driven by genetic differences. Simultaneously, non-genetic factors may account for twin discordance in features of FASD or the diagnosis itself, such as differences in placentation, rates of organogenesis, and fetal vasculature. These aspects need to be more thoroughly explored. There has been at least one early case study in 1974 showing that MZ twins had “slightly different physical anomalies”, suggesting that subtle differences in the development of MZ twins may alter alcohol teratogenicity [27].

Twin studies and studies of other pairs of individuals at differing degrees of relationship can be confounded by ascertainment bias, with pairs of children with FASD perhaps being more likely to be recognized. However, as far as is known, this is not the case for the heritability of FASD, leaving the conclusion that FASD is highly heritable, although more influenced by environmental factors in different aspects that lie within the spectrum of FASD. Discordant diagnosis in DZ twins also provides evidence for the significance of offspring genotype in PAE vulnerability rather than the maternal genotype; however, as discussed in the next sections, maternal genotype can also be a significant factor in FASD vulnerability. While these studies have provided evidence for a genetic contribution to FASD, only two studies thus far have included both MZ and DZ twins, with only 5 and 9 MZ twin pairs and only 11 and 39 DZ twin pairs. Thus, further studies and larger cohorts are needed to validate these findings, particularly including more MZ twins.

### 2.1. Alcohol-Metabolizing Enzymes

As has long been recognized and more recently confirmed in genome-wide association studies (GWASs) [28], genetic polymorphisms influencing alcohol drinking and thus the propensity for alcohol use disorder (AUD) occur at enzymes for metabolizing alcohol, in particular alcohol dehydrogenase (ADH) and acetaldehyde dehydrogenase (ALDH). ADH metabolizes alcohol to acetaldehyde which is then metabolized to acetate by ALDH. Several distinct classes of human ADH and ALDH isozymes exist, and remarkably, in the context of the genetics of behavior, common genetic variants in at least two of the genes are common, functional, and of large effect on the metabolism of alcohol, consumption of alcohol, and consequences of alcohol intake. These two polymorphisms within the *ADH1B* gene are Arg48His (rs1229984, *ADH1B*2*) and Arg370Cys (rs2066702, *ADH1B*3*) [29]. These allelic isoforms have different alcohol-metabolizing rates, with *ADH1B* His48 and *ADH1B* Cys370 metabolizing ethanol to acetaldehyde more than 70–80 times faster [30,31]. These polymorphisms vary dramatically in frequency across populations and, as far as is known, are unique to humans. *ADH1B* His48 has an allele frequency greater than 60% in Chinese, Japanese, and Korean populations [32], and *ADH1B* Cys370 has an allele frequency of approximately 22–25% in African Americans [33]. On the other hand, *ADH1B* Arg48 and Arg370 (*ADH1B*1*) are predominant in European white individuals. In addition to these common *ADH* variants, the Lys504 allele of the Glu504Lys *ALDH2* polymorphism encodes a low-activity variant of the enzyme and is present in up to half of people in certain East Asian populations [34]. People with *ADH1B* His48, *ALDH2* Lys504, or both are less likely to develop AUD because of the flushing reaction caused by the acetaldehyde-induced release of histamine. Paradoxically, these individuals are also more vulnerable to the toxic effects of alcohol, including DNA damage and cancer, because many individuals with the Asian flush do consume at least moderate amounts of alcohol [35]. This vulnerability is now a greater public health concern and is possibly also germane to the teratogenic effects of alcohol exposure because of the widespread use of antihistamines and other drugs that block the flush but not the carcinogenicity of acetaldehyde.

It is perhaps unsurprising that alcohol metabolic gene variants have also been associated with differential teratogenicity of PAE (Table 1). Partly, this may be because binge drinking resulting in high peak alcohol concentration is a greater risk factor for alcohol teratogenicity than the daily average quantity [36]. In addition to the pattern of alcohol intake, the metabolic activity of the mother also determines peak alcohol exposure; thus, the differences in alcohol metabolism can contribute to variable risks and outcomes after PAE. The contribution of offspring *ADH* genotype in FASD vulnerability is dependent on the developmental timing of gene expression, with *ADH1A* being expressed in the early fetal liver, *ADH1B* being detectable after the first trimester, and *ADH1C* being expressed in the liver after birth [37]. Furthermore, although expressed, the activity of alcohol-metabolizing enzymes does not reach adult levels until early childhood [38]; therefore, the contribution of variation in alcohol-metabolizing genes is expected to be more relevant for the mother as opposed to the offspring, which largely relies on the mother for eliminating alcohol.

Evidence that the common *ADH* variants contribute to FASD susceptibility has been accumulated from several populations. A study on African American women at the University Health Center in Detroit, Michigan stratified subjects by maternal alcohol intake levels and *ADH1B* genotype [39]. In the 243 mother–infant pairs, the *ADH1B* Cys370 (*ADH1B*3*) allele was associated with faster development in the offspring of drinking mothers but not non-drinking mothers, with the offspring of Cys370 carriers being developmentally more like offspring of non-drinking women. The protective effects of the Cys370 allele are thought to be related to its higher V_max_ for alcohol metabolism, thus decreasing peak alcohol levels reached in these carriers and also suggesting that alcohol, rather than acetaldehyde, acts as the prominent teratogen. However, it could also be due to the tendency of Cys370 carriers to flush, thus reducing or altering the pattern of alcohol consumption. Because the study enrolled participants with minimal differences in periconceptional alcohol intake within each genotype, differences in alcohol intake by genotype should not account for the differences in offspring outcomes. Interestingly, offspring with the *ADH1B* Cys370 allele similarly showed improved development, an unexpected finding given the low activity of ADH during gestation. The same group later investigated whether the *ADH1B* genotype influenced facial morphology in the same cohort of patients. By analyzing photographs of 247 offspring of mothers with a range of alcohol use during pregnancy, researchers found the Cys370 allele, whether in mother, offspring, or both, had a protective effect against alcohol-induced facial dysmorphology [40].

The potential long-term effects of the *ADH* genotype on FASD were later studied in the Detroit Longitudinal Prenatal Alcohol Exposure Cohort, where offspring were assessed at infancy and 7.5 years. This study of a cohort of 263 mother–child pairs of African ancestry found that mothers carrying at least one copy of the Cys370 allele had protective effects on birth size as well as behavioral and cognitive outcomes at both time points. In contrast with the previously mentioned study, there was no consistent effect on the infant genotype [41]. Alternatively, children whose mothers were Arg370/Arg370 homozygotes had increased social and behavioral problems after PAE, and women lacking the Cys370 allele drank twice as much as those with at least one copy of the allele. A follow-up study investigating alcohol-related behavioral issues in adolescence was later conducted at 14 years. PAE overall was associated with aggression and externalizing behavioral problems. In adolescents with mothers lacking the Cys370 allele, hyperactivity and inattention were more common [42].

Studies from other populations have also shown trends for *ADH* genotypes altering risk for FASD. In a study of 56 mothers and their FAS children in a population of Khoisan–Caucasian mixed-ancestry women in the Western Cape Province of South Africa, the allele frequency of *ADH1B* His48 (*ADH1B*2*) was lower in FAS children and mothers [44]. These findings align with previous studies to show that the *ADH1B* His48 allele is associated with a lower frequency of drinking. Although *ADH1B* Cys370 was shown to be protective against FASD in studies of African American women in the Detroit studies [39,40,41], the low frequency of this allele in this mixed-ancestry population (0.039 for controls and 0.036 in affected) suggests its relatively lower impact on FAS risk. Another study of *ADH*/*ALDH* genotypes in a mixed-race population was performed on 404 high-risk pregnant women and 139 infants in the Boston, Massachusetts, area. In this study, mothers who were *ADH1B* Cys370 carriers drank more and had an increased risk for having children with growth restriction and/or FAS facial features [43], and 60% of affected infants were carriers compared to only 29% of infants who were unaffected. These findings contradict the Detroit studies, in which *ADH1B* Cys370 conferred protection against FASD. The authors attributed this discrepancy to differences in the phenotypes measured between studies, but it may instead be due to differences in timing and quantity of drinking by the mothers in these two studies conducted at different times and in different populations.

Further studies have been conducted from the Avon Longitudinal Study of Parents and Children (ALSPAC), a population-based study in England investigating the effect of environmental factors on child health and development. In a Mendelian randomization study of 7410 white women, polymorphisms in four ADH genes predicted patterns of maternal alcohol intake. One non-synonymous variant, *ADH1B* rs1229984, a rare variant associated with accelerated ethanol clearance, was associated with lower alcohol consumption during pregnancy and a higher likelihood to abstain from alcohol use altogether [45]. The role of the *ADH* genotype in the effects of moderate alcohol drinking on cognitive and behavioral outcomes in offspring was also investigated in the ALSPAC cohort. In an analysis of ten single nucleotide polymorphisms (SNPs) in four ADH genes, four variants were related to differences in IQ present in children of mothers who drank moderately during pregnancy [46]. The predictive value of the genetic variants for child IQ was more significant for the child’s genotype as compared to the mother, suggesting a role of fetal alcohol metabolism in FASD risk. A Mendelian randomization study in 3500 children of white ethnicity from the ALSPAC showed that variation in the child’s alcohol-metabolizing genes predicted increased risk for early onset persistent conduct problems in children of mothers who engaged in moderate drinking during pregnancy but not in children with conduct problems born to non-drinking mothers [47]. The ADH-related genes had no predictive value for conduct issues that were childhood-limited or adolescence-onset, supporting hypotheses that conduct problems of different developmental trajectories may have different etiologies, some potentially being more specific to PAE.

Further studies are warranted to elucidate the role of ADH genes in FASD vulnerability. Thus far, studies are limited to relatively small sample sizes in specific individuals of particular demographics. Although conflicting evidence exists, *ADH* variants which accelerate alcohol metabolism appear to have a protective effect against FASD. This could be due to differences in levels and developmental alcohol consumption, although some of the studies at least controlled for level of exposure. The protective effect of *ADH* alleles could also be due to a decrease in peak alcohol levels, implicating ethanol rather than acetaldehyde as the primary teratogenic agent. Despite the well-known toxicity and carcinogenicity of acetaldehyde, studies in mice support the idea that ethanol is the primary teratogen in FASD, showing that inhibiting ADH activity, but not ALDH, increases alcohol teratogenicity [53,54]. Direct administration of acetaldehyde, however, has been shown to be teratogenic in both mice and zebrafish [54,55]. Thus, acetaldehyde is also likely to contribute to FASD. Interestingly, the effect of offspring *ADH* genotype has been shown to confer differential vulnerability in some but not all of the studies. The contribution of fetal metabolism of ethanol is estimated to be only 3–4% of the maternal rate [56]; therefore, how fetal *ADH* variants lead to protection or vulnerability is unknown, but, as was observed earlier, the *ADH1B* gene is not expressed early in gestation, and thereby fetal *ADH1B* genotype might well be serving as a surrogate for maternal genotype and alcohol metabolic capacity, at least at early developmental timepoints. While initially low, the fetal capacity to metabolize alcohol does reach around 84% in the newborn [57], and differences in the effect of offspring *ADH* genotype may be dependent on the timing of exposure, with fetal genotype expected to be a greater factor at later stages in development. The relationship of *ADH* polymorphisms to FASD risk requires further investigation of fetal and maternal genotypes in the context of the timing of alcohol exposure and may be more feasible in humanized genetic animal models.

### 2.2. Other Genes and Pathways

Beyond polymorphisms altering alcohol metabolism, and still in advance of genome-wide studies, some other genetic variants involved in general developmental processes or alcohol use have been implicated in FASD (Table 1). The evidence supporting these genes is at this point not as compelling as is the role of alcohol metabolic gene variants that also powerfully predict carcinogenicity and AUD itself, and it will be of interest to see whether these genes, attractive as they may be on the basis of their developmental roles, exhibit effects when examined in genome-wide contexts.

Associations of both maternal and offspring genes related to development have been made in a few studies, albeit with small sample sizes. These genes include *CYP17*, which encodes cytochrome P450c17α which acts in estrogen biosynthesis. Mutations in *CYP17* are associated with maternal hormone imbalance and intrauterine growth restriction (IUGR), the latter being a commonly reported feature of FASD. In a hospital-based case-control study in Liverpool comparing low to normal birthweight pregnancies (N = 180 controls and 90 cases), a polymorphism in the untranslated region of *CYP17* was more frequent in mothers of babies with IUGR if they had consumed alcohol during pregnancy [48], suggesting a gene by alcohol interaction for fetal growth. *IGF2/H19* is a parentally imprinted gene critical for normal placental and embryonic growth and has been previously implicated in altered DNA methylation after PAE in both mice and humans [49]. An SNP in *CTCF6* (rs10732516), a zinc finger protein that regulates *IGF2,* was previously implicated in newborn and adult height [58]. In placentas from 39 PAE and 100 control newborns, *CTCF6* rs10732516 was associated with both a decrease in newborn head circumference (HC) and decreased methylation at the H19 imprinting coding region (ICR) of *IGF2* in alcohol-exposed placentas [49]. Only 14 of the 44 PAE placentas carried the rarer allele; therefore, larger sample sizes are needed to further support the roles of *IGF2* and *CTCF6* in PAE-induced reductions in HC. Finally, variants affecting N-glycosylation have also been associated with an increased risk of FAS. FAS has several overlapping clinical presentations with congenital disorders of glycosylation, and in a study of 25 FAS patients and 20 controls, rare variants in genes associated with N-glycosylation presented at a higher frequency in patients, suggesting these variants might increase the risk for FAS [51]. Further research on genes modulating the post-translational addition of carbohydrate moieties to proteins, and the consequences of disturbances of these mechanisms, is merited in FASD.

Whether genetic variants might impact the effectiveness of FASD interventions has also been investigated. Dietary choline intake during pregnancy has been shown to improve memory, cognition, and attention in young children, and choline supplementation has been shown to reduce some deficits consequent to PAE [50]. Fifty-two children 2 to 5 years of age diagnosed with FASD received 6 months of placebo or choline supplementation. Associations with 14 SNPs in the choline transporter gene *SLC44A1* out of 243 tested SNPs associated with choline were significantly associated with increased cognitive performance [50]. These SNPs or functional variants in linkage disequilibrium with them may partially explain the variable effects of choline supplementation in children with FASD.

In addition to genes that influence development, studies have also investigated genes associated with alcohol use in general. The Danish National Birth Cohort (DNCB) examined the extent to which candidate genes for smoking, alcohol use, and obesity predicted the use of these substances as well as weight gain during pregnancy. A relatively small GWAS of 1937 mothers revealed associations at 39 genes including prodynorphin (*PDYN*), *ADH1C*, and the glutamate metabotropic receptor (*GRM8*) and drinking behaviors at different stages of pregnancy [52]. None of the identified SNPs, however, were significantly associated, possibly because of limited sample size against the likely small effect of any individual locus. However, it is likely that genes altering alcohol consumption in mothers overlap at least partly with genes detected in much larger GWAS cohorts. In those GWASs, numerous genome-wide significant signals have been found (including *ADH1B* and *ALDH2* as previously mentioned), and polygenic scores (PGSs) derived from them presently account for up to 5% of the variation in AUD [59]. Testing the effects of these polygenic scores on PAE is warranted, as is the collection of larger samples of mothers in whom PAE is assessed.

### 2.3. Identification of Copy Number Variations by Genetic Testing

As several of the conditions associated with FASD are also common to other neurodevelopmental disorders seen by pediatricians and neonatologists, it is not uncommon for clinical geneticists to be consulted for potential FASD and to delineate FASD from other causes. The British Medical Association recommends genetic testing for all children with suspected FASD [60]. The American College of Medical Genetics and Genomics recommends chromosomal microarray analysis (CMA) as the first-tier genetic test for children with developmental delay and/or other congenital abnormalities, as submicroscopic chromosomal alterations are estimated to occur in 10–15% of these patients [61]. Past studies support the use of genetic testing in which children with suspected FASD may have an alternative genetic disorder. Of 80 patients referred to the Manchester Genetic Services for possible FASD, only 42 were given an FASD diagnosis, with 6 of the non-FASD cases revealing pathogenic microdeletion/duplication syndromes [60]. In 27 children with suspected FASD referred to the Pediatric Genetics Department at the Academic Medical Center in Amsterdam, 16 children did not meet the criteria for FASD, and 1 of those 16 children had a deletion, and another had a duplication [62].

Genetic testing has also revealed chromosomal alterations in patients with FASD diagnosis. A retrospective study of 36 patients referred for genetic testing at Boston Children’s Hospital found that 8 of the referred patients did not meet the criteria for FASD and were given an alternative diagnosis, with the remaining having either full FAS, partial FAS, or ARND [63]. A CMA was performed in 25 of the FASD patients, with 3 revealing pathogenic CNVs. The genetic syndromes caused by these CNVs overlapped phenotypically with FASD. In a genome-wide analysis of CNVs in 95 FASD children and 87 aged-matched controls from the NeuroDevNet Canadian FASD study cohort, rare CNVs were detected in 90% of cases with FASD, with around half of those impacting coding regions [64]. Fourteen of the identified CNVs in 12 of the individuals have been previously implicated in brain function, neurodevelopment, or other genomic disorders. In a review of 110 cases at the Yorkshire Regional Genetics Service from 2013 to 2017 coded with “maternal alcoholism” or “foetal alcohol”, only 4 (3.6%) were found to have genetic abnormalities in this analysis, with all being chromosomal abnormalities [65]; however, the cases in this study did not have a confirmed FASD diagnosis.

Beyond these cohort studies that indicate the potential for interaction between PAE and CNVs, a follow-up report was also performed on discordant DZ twins from the 1994 study by Riikonen to explore the possible genetic contributions of differential vulnerability to the effects of PAE. Using microarray-based comparative genomic hybridization (aCGH), a microdeletion at 18q12.3-q21.1 was detected in the affected twin [66]. Deletions in chromosome 18 are typically de novo mutations and result in what is called 18q deletion syndrome, with phenotypes that overlap with FASD, including intellectual disability, short stature, and facial dysmorphism. Whether the microdeletion caused separate effects from PAE and/or exacerbated effects of PAE is unknown. Because the twin pair had confirmed PAE and a full FAS diagnosis, including the characteristic facial features specific to PAE, alcohol teratogenicity can be assumed regardless of the presence of the microdeletion. The 18q microdeletion is thought to be a common de novo CNV, suggesting that mutations of this nature may account for some proportion of FASD vulnerability among twins and the general population. Indeed, studies of healthy individuals have shown that CNVs are common throughout the human genome [67] and contribute to neurodevelopmental disorders. As described here, increased genetic testing for CNVs may not only assist in diagnosis but also uncover sources of genetic vulnerability to FASD.

## 3. Discussion

In this review, we summarized findings investigating potential genetic influences on FASD. Genetic variation both within the mother and the infant may alter FASD, maternal genetic variation perturbing alcohol metabolism and alcohol intake, both of which largely determine fetal alcohol exposure, as well as variation related to the uterine environment. Fetal genotype can also alter alcohol metabolism and, critically, a variety of fetal genetic variations can either directly cause malformations resembling FASD or, as is likely, interact with alcohol exposure to mediate differences in teratogenicity of alcohol. Early twin case studies first established the potential link between genetic factors and PAE outcomes, with several cases of DZ twins presenting discordant FASD diagnosis, despite virtually equivalent PAE and uterine environments. Later twin studies, which included MZ twins, further support a genetic component of FASD, with MZ twins showing 100% concordance, reflecting a remarkably high heritability that still needs to be confirmed via larger studies of twins and other pairs of relatives, or by identification of genes responsible. The effect of common variants for the alcohol-metabolizing enzyme ADH has been explored in populations of different ancestries, with different variants showing some protective effects in different populations. Candidate gene studies in smaller cohorts have linked other variants related to growth and nutrient metabolism to FASD outcome, and polymorphisms linked to AUD showed some associations with drinking behaviors surrounding pregnancy in a GWAS study. Genetic testing for FASD has revealed CNVs that were both associated with the condition, including a microdeletion found in one affected twin in a discordance DZ pair, and also those that potentially explain phenotypes in children incorrectly diagnosed with FASD. Overall, these studies provide some gene-based evidence that heritability influences the variability in FASD and may partially explain why not all children with PAE manifest FASD phenotypes.

As mentioned earlier in this review, a significant limitation in the interpretation of existing studies is the small sample sizes. Twin studies included only 5 and 9 MZ twins and 11 and 39 DZ twins, and the candidate genes ranged from 45 to 404 mother–offspring dyads. Studies performed in the Avon Longitudinal Study of Parents and Children (ALSPAC) and The Danish National Birth Cohort (DNCB) had sample sizes of 1900 to 7400, which while an improvement is still relatively small in the era of large-cohort genomic studies. For example, GWA studies of height and body mass, far easier phenotypes to assess, are conducted on greater than one million individuals. Given the known impact of ancestry, particularly for the *ADH* variants, a wide range of drinking patterns during pregnancy, and a variety of maternal factors which influence FASD, large, deeply phenotype longitudinal studies will be ultimately needed to isolate the effects of individual genes and to identify the cumulative, polygenic contributions of many genes that may be combined into polygenic risk scores and used both in research and, one day, clinically. Importantly, this would also facilitate the identification of novel genes and pathways to FASD which are largely unknown at this point.

Both determining diagnosis and elucidating the etiology of FASD are complicated by the number of associated comorbid conditions. A recent study found 428 comorbid conditions with FASD, with the most commonly occurring conditions being related to congenital malformations, deformities, and chromosomal abnormalities [7]. This includes both trisomy 21 and ring chromosome 6, which have been associated with FASD in previous studies [68,69,70]. One of the most common phenotypes associated with FASD is attention deficit, and the prevalence of attention-deficit hyperactivity disorder (ADHD) in children with FASD is as much as 10-fold elevated [71]. As ADHD is relatively common (10% of the general population) and highly heritable (77%), and children diagnosed with ADHD have more alcohol-related problems in adulthood [72,73,74], it is possible that PAE may increase the susceptibility for ADHD in individuals already vulnerable to the disorder. Thus, the interaction of developmental alcohol exposure and vulnerabilities for other disorders may account for a large portion of these comorbidities.

Given the wide range of phenotypic outcomes of PAE as well as the number of factors that contribute to severity, understanding the potential genetic contributions to FASD will likely need to focus on specific phenotypes (i.e., facial morphology or attention issues) within specific populations (i.e., people of particular ancestries and cultures) and account for the different factors that influence developmental outcomes (i.e., maternal stress or malnutrition). GWASs and other genomic studies may validate but will hopefully also expand our knowledge of molecular pathways to FASD, for which there are now many theories. In addition to further genetic and genomic studies in larger populations for identifying risk or resilience variants for FASD, findings across different teratogenic syndromes and neurodevelopmental disorders as well as complementary non-human models will continue to be fundamental in understanding the genetic contributions to PAE phenotypes and FASD risk factors. FASD represents a complex interplay of environments, including the environment of the mother, the environment within the mother, and the environmental exposure to alcohol. Understanding the interaction of both maternal and offspring genes within these environments across the trajectory of development poses a challenge for scientists and clinicians in identifying genetic factors that might confer different vulnerabilities. As there is no cure for FASD, identifying these genetic vulnerabilities could aid in early detection and intervention preventing FASD or mitigating the consequences of PAE.

## Figures and Tables

**Table 1 genes-14-00195-t001:** Polymorphisms associated with PAE and FASD.

Polymorphism	N	Findings	Source
Alcohol dehydrogenase (ADH)			
*AHD1B* Cys370 (ADH1B*3)	247 African American mother–offspring pairs	Improved development in infants of drinking mothers carrying Cys370 Protective effect of Cys370 against PAE-induced facial dysmorphology	McCarver et al., 1997 [39] Das et al., 2004 [40]
	263 African American mother–offspring pairs	Protective effects of Cys370 on birth size as well as behavioral and cognitive outcomes at infancy and 7.5 years Cys370 carriers were less likely to have hyperactivity and inattention at 14 years	Jacobson et al., 2006 [41] Dodge et al., 2014 [42]
	404 mixed-race women and 139 infants	Cys370 carrier mothers had increased risk for growth restriction and/or facial dysmorphia; Cys370 was more frequent in affected infants	Stoler et al., 2002 [43]
*ADH1B* His48 (ADH1B*2)	56 mixed-ancestry mother–offspring pairs	Lower frequency of His48 in FAS children and their mothers	Viljoen, et al., 2001 [44]
*ADH1B* rs1229984	7410 white European women	Associated with alcohol consumption during pregnancy	Zuccolo et al., 2009 [45]
*ADH1A* rs2866151 *ADH1A* rs975833 *ADH1B* rs4147536 *ADH7* rs284779	6196 children of white European women	These 4 ADH loci of 10 tested were associated with lower IQ in children of mothers who drank moderately during pregnancy	Lewis et al., 2012 [46]
	3500 children of white European women	The same 4 ADH loci predicted increased risk of early-onset persistent conduct problems among children of mothers who drank moderately during pregnancy	Murray et al., 2016 [47]
Other Genes			
*CYP17 A1 allele*	180 controls and 90 cases, Caucasian women	Associated with intrauterine growth restriction in infants of mothers who consumed alcohol during pregnancy	Delpisheh et al., 2008 [48]
*CTCF6* rs10732516	100 Caucasian controls and 39 predominately Caucasian alcohol-exposed	Associated with decreased newborn head circumference and decreased methylation at the H19 imprinting coding region (ICR) of *IGF2* in alcohol-exposed placentas	Marjonen et al., 2017 [49]
*SLC44A1*	52 children diagnosed with FASD	14 SNPs in *SLC44A1* were associated with increased cognitive performance after choline supplementation	Smith et al., 2021 [50]
N-glycosylation genes	20 controls and 25 FAS patients of different European ancestries	Rare variants in genes associated with N-glycosylation were more frequent in FAS patients	de la Morena-Barrio et al., 2018 [51]
*PYDN*, *ADH1C*, and *GRM8*	1937 mothers of primarily Caucasian background	SNPs in 39 genes (including *PYDN*, *ADH1C*, and *GRM8*) were associated with drinking during different stages of pregnancy	Wehby et al., 2015 [52]

## Data Availability

Not applicable.

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
