# Peer review of "Genetic Influences on Fetal Alcohol Spectrum Disorder"

_genes, 2023, doi:10.3390/genes14010195_

Round 1

Reviewer 1 Report

This paper represents a considerable amount of work and careful analysis of the genetics of Fetal alcohol spectrum disorder (FASD). The subject is well introduced providing the reader sufficient background and discussion.

The authors should show more clearly what is the objective and advantage of their review in comparison of the other recent reviews in this topic ( PMID: 27122355;   PMID: 28820871;  PMID: 24917878 )   Optional: Figure 1 would look more scientific if it would be cropped from the bottom I have no other comments about the paper.  

Author Response

We thank the Reviewer for the suggestions and have commented in the introduction about how this review is different from those previously published.

Reviewer 2 Report

The authors provide a narrative review on genetic influences on fetal alcohol spectrum disorder. The topic is important and interesting, and the review stimulates future research directions. Here are my comments:

To facilitate readability and to provide a better overview, I suggest to include tables that list the referenced studies with publication year, the number of cases included, and the major finding(s).

The manuscript would strongly benefit from a separate and more focused paragraph on the limitations of the available literature and the resulting requirements for future studies.

Introduction: “While FAS does not require documented PAE, partial FAS (pFAS) can be diagnosed with confirmed PAE and features A and B, or without confirmed PAE and features A, B, and C.” The authors should verify the correct assignment of features. For example, required features for pFAS with PAE are facial anomalies and neurobehavioral impairment (which are A and D here) according to Hoyme et al. 2016.

Author Response

We thank the Reviewer for the helpful comments. As suggested, we have now included a table in the review to summarize the studies described. We have also added a paragraph in the Discussion describing current limitations in the field. Finally, we have revised the section of the Introduction regarding the FAS features.